# Steering Risk Preferences in Large Language Models by Aligning Behavioral and Neural Representations

## Abstract

Changing the behavior of large language models (LLMs) can be as straightforward as editing the Transformer's residual streams using appropriately constructed "steering vectors." These modifications to internal neural activations, a form of representation engineering, offer an effective and targeted means of influencing model behavior without retraining or fine-tuning the model. But how can such steering vectors be systematically identified? We propose a principled approach, which we call *self-alignment*, that uncovers steering vectors by aligning latent representations elicited through behavioral methods (specifically, Markov chain Monte Carlo with LLMs) with their neural counterparts. To evaluate this approach, we focus on extracting latent risk preferences from LLMs and steering their risk-related outputs using the aligned representations as steering vectors. We show that the resulting steering vectors successfully and reliably modulate LLM outputs in line with the targeted behavior.

## 1 Introduction

LLMs are increasingly deployed in risk-sensitive domains such as finance (Niszczota & Abbas, 2023) and healthcare (Shmatko et al., 2025). In these high-stakes applications, it is essential to develop reliable methods for aligning the behavior of LLMs with human values and safety requirements (Hendrycks, 2025; Mazeika et al., 2025). One solution to this problem is *steering*, a term that refers to any targeted intervention (whether through model weights, decoding strategies, prompts, or internal neural activations) intended to control or shape a model's outputs. Steering risk-related behavior in LLMs is one way to ensure alignment with humans in risky domains.

Steering the risk preferences of a pretrained LLM, however, is inherently challenging due to the opacity of these models. LLMs operate over vast weight spaces, and their outputs are highly context-dependent (Brown et al., 2020; Zhu & Griffiths, 2024b), making it difficult to isolate or manipulate any specific internal variable that governs risk preference. Existing steering techniques, such as prompt engineering or supervised fine-tuning, either lack the granularity needed to target specific latent representation or require extensive retraining and human supervision (Qi et al., 2023; Ziegler et al., 2019).

Given the context-dependent and probabilistic nature of LLM outputs, we propose that their underlying risk preferences are best characterized as *probabilistic representations*. That is, the same risky decision may yield different completions depending on subtle variations in input phrasing or prompt structure (Zhu & Griffiths, 2024b; Brown et al., 2020). This view suggests that an LLM's underlying risk preferences can be recovered by sampling its behavior over many such comparisons. To operationalize this idea, we measure the risk preferences of LLMs using a method based on Markov chain Monte Carlo (MCMC), in which repeated choices made by the model define a Markov chain that converges to a probability distribution representing these preferences. This approach has previously been used to elicit probabilistic representations in other settings from both humans and LLMs (Noguchi et al., 2013; Sanborn & Griffiths, 2007; Harrison et al., 2020; Zhu et al., 2024).

Measuring the risk preferences of LLMs in this way creates the opportunity to build a bridge between the observed behavior and the activations of nodes in the underlying neural network (Sucholutsky et al., 2023). We can align the behavioral representations we find with the neural representations

within LLMs and use this alignment to derive a steering vector that captures underlying risk preferences. When this vector is injected back into the model at inference time, it enables precise control of the model's risk-related behavior. We refer to this approach as *self-alignment*, since it leverages the model's own emergent representations for behavior control.

To evaluate this method, we apply steering vectors derived from the aligned representation across three domains: (i) risky decision-making, (ii) risk perception, and (iii) risk-related text generation. Our results demonstrate that self-alignment yields substantially greater control over model behavior than the alternative Contrastive Activation (CA) approach (Panickssery et al., 2023; Turner et al., 2023), which derives steering vectors from prompt pairs. Our results also demonstrate that the modified risk preferences transfer to tasks that are quite far away from the choices from which the steering vectors were derived.

## 2 BACKGROUND

**AI safety and value alignment.** AI safety research has long emphasized the importance of aligning artificial systems with human values, though explicitly encoding such values in machines remains a formidable challenge (Russell, 2022). The emergence of LLMs presents new opportunities in this regard, as LLMs have been shown to internalize commonsense knowledge and human norms from large-scale training data (Hendrycks et al., 2020; Mazeika et al., 2025; Marjieh et al., 2024). Consequently, many researchers now argue that, given sufficient data, LLMs can approximate shared norms (Brown et al., 2020). At the same time, however, this capacity introduces new challenges for interpretability, as it becomes increasingly difficult to discern the underlying factors that drive LLM behavior. In this work, we propose a novel strategy that leverages emergent representations in LLMs by eliciting the same latent construct through both behavioral and neural means. This dual elicitation facilitates self-alignment between behavior and neural activations, providing an interpretable method for steering LLM outputs.

**LLM steering.** A range of approaches has been proposed to influence the outputs of pretrained LLMs, which can broadly be categorized as different forms of intervention. These interventions vary in where and how they modify the model. Weight-level interventions include techniques such as supervised fine-tuning (Qi et al., 2023) and reinforcement learning from human feedback (Ziegler et al., 2019), which directly update model parameters. Alternatively, decoding-level interventions, such as trainable decoding, modify the output generation process while keeping model weights fixed (Grover et al., 2019). Prompt engineering can be viewed as an intervention on the input space, shaping model behavior through carefully constructed prompts (Zhou et al., 2022; Yao et al., 2023). Moreover, activation-level interventions, which typically freeze the model weights and instead search for steering vectors, offer an alternative to behavioral control. These vectors can be discovered through gradient-based optimization (Hernandez et al., 2023) or computed directly from contrastive prompt pairs (Li et al., 2023; Turner et al., 2023). In this work, we propose an alignment-based method for deriving steering vectors by aligning behavioral and neural representations of latent constructs such as risk preference.

## 3 METHOD

The key idea behind our proposed method is to derive a steering vector that optimally aligns the model's behavioral and neural representations of risk preference (see Figure 1a). Our method proceeds in two main steps, described below.

**Step 1: Eliciting behavioral representations of risk via MCMC.** Risk preference, like many other mental representations, is inherently unobservable. However, as demonstrated in cognitive psychology, such latent constructs can be inferred from observed behavior (Kay & Cook, 2023; Sanborn & Griffiths, 2007; Harrison et al., 2020). Drawing inspiration from recent work on behavioral elicitation in LLMs (Zhu & Griffiths, 2024a; Zhu et al., 2024; Capstick et al., 2024), we incorporate LLMs into a MCMC sampler to effectively elicit their behavioral representation of risk. A variety of sampling algorithms can be used for this purpose, including the Metropolis-Hastings algorithm (Zhu et al., 2024; Sanborn & Griffiths, 2007) and Gibbs sampling (Harrison et al., 2020). The core intuition is to use the LLM to define the proposal or acceptance mechanism, such that the resulting sequence of samples produced by the Markov chain converges to a stationary distribution that

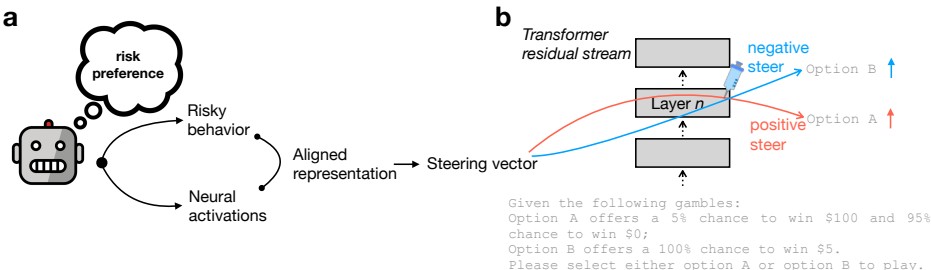

Figure 1: **Self-aligned steering vectors. (a)** Overview of the proposed method for generating steering vectors by aligning representations of risk preference derived from behavioral and neural elicitation. **(b)** During inference, the steering vector is injected into the residual stream at all token positions to control LLM outputs. When the steering vector is applied with a positive multiplier (i.e., positive steering), the LLM is expected to exhibit more risk-seeking behavior. Conversely, applying a negative multiplier (i.e., negative steering) is expected to induce more risk-averse behavior.

reflects the model's latent representations (Zhu et al., 2024; Noguchi et al., 2013; Harrison et al., 2020; León-Villagrá et al., 2020; Yan et al., 2024; Sanborn & Griffiths, 2007).

Specifically, we adapted the procedure established by Noguchi et al. (2013), which has been successfully used to elicit human risk preferences, to the LLM setting. In each trial, the LLM is prompted to choose between two gambles, A and B. In our implementation, all gambles consist of three possible outcomes: $0, $50, and $100. While the outcome values are fixed, the probabilities associated with each outcome vary across gambles. After the LLM makes a choice, the selected gamble is retained, and the unchosen option is replaced with a newly generated gamble. The probabilities for this new gamble are randomly sampled from a Dirichlet distribution: $\mathrm{Dir}(1, 1, 1)$. In the next trial, the retained and newly generated gambles are presented again (with their order randomized), and the LLM is asked to make another choice. Importantly, no choice history is provided in the prompt: each decision is made solely based on the current pair of gambles presented. The sequence of choices made by the LLM forms the foundation for constructing its behavioral representation of risk. Specifically, within the space of all possible gambles, represented as a probability triangle[1], the LLM's choices allow us to infer a probability distribution over gambles reflecting its preferences across risk profiles (see Figure 2b).

More formally, MCMC begins at an initial state $z$ (i.e., a specific gamble from the probability triangle). A proposed next state $z'$ is drawn from a proposal distribution $q(z'|z)$, and is then evaluated under the target distribution $\pi$ (i.e., the LLM's latent representation of risk) to determine whether it should be accepted as the new state or rejected in favor of retaining the current state $z$. To guarantee that the Markov chain converges to $\pi$, it is sufficient to satisfy the condition of detailed balance (along with ergodicity):

$$\pi(z)q(z'|z)A(z', z) = \pi(z')q(z|z')A(z, z') \tag{1}$$

where $q(z'|z)$ is the probability of proposing $z'$ from state $z$, and $A(z', z)$ is the probability of accepting proposal $z'$ over $z$. In our case, we use a symmetric proposal distribution (i.e., $q(z'|z) = q(z|z')$), which simplifies the detailed balance condition to: $\pi(z)A(z', z) = \pi(z')A(z, z')$. One way to satisfy this condition is by using the Barker acceptance function (Barker, 1965):

$$A(z', z) = \frac{\pi(z')}{\pi(z) + \pi(z')} \tag{2}$$

This acceptance rule is particularly appropriate for modeling LLM behavior, as it closely resembles well-known stochastic choice models such as Luce's choice rule (Luce et al., 1959) and the

---

[1]In the economics literature, this triangle is also known as the Marschak–Machina probability triangle (Marschak, 1950; Machina, 1982), a method traditionally used to qualitatively differentiate among competing theories of risky choice (Wu & Gonzalez, 1998). Our MCMC approach basically provides a non-parametric estimation of the LLM's risk representation within this triangle.

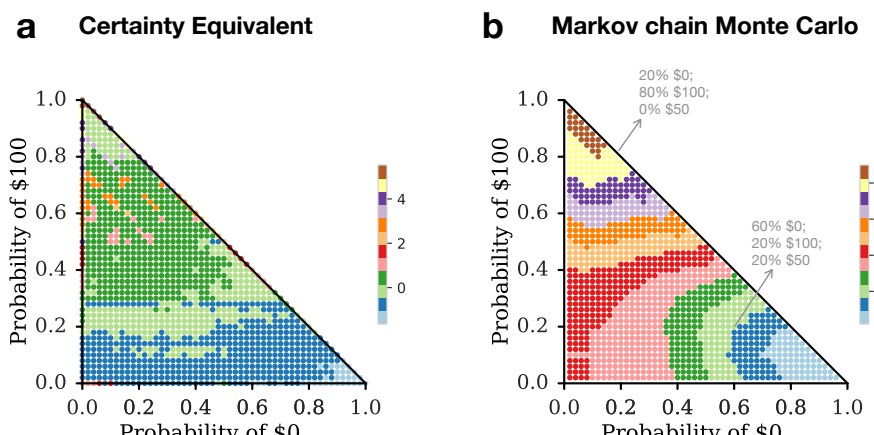

Figure 2: **Elicited risk preferences from Gemma-2-9B-Instruct using behavioral methods. (a)** Certainty Equivalent method. **(b)** Markov chain Monte Carlo with LLM. Each triangle represents the probability simplex over three-outcome gambles ($0, $50, and $100), where the sum of outcome probabilities equals one. The MCMC-with-LLM elicitation reveals more nuanced and structured contours of risk preference compared to the Certainty Equivalent method. Higher values indicate a stronger preference for the gamble by the Gemma model.

Bradley–Terry model (Rafailov et al., 2023), which has been applied in LLM post-training and preference alignment. As a result, by sequentially presenting pairs of risky choice alternatives to an LLM, the set of selected options can be interpreted as samples from a probability distribution whose density is proportional to the LLM's latent representation of risk.

**Step 2: Aligning behavioral and neural representations to compute steering vectors.** Given the behavioral representations of risk elicited via MCMC with LLM, we next sought to align them with the model's internal neural activations. To obtain corresponding neural representations of risk preference for each gamble within the triangle, we prompted the same LLM to evaluate the attractiveness of the gamble when hypothetically offered (see Appendix A.4 for details). We then aligned the two representations by regressing the behavioral estimates onto the neural activations. Specifically, we treated the neural activations as independent variables and the behavioral responses as dependent variables, using Lasso regression.

More formally, the LLM provided direct valuations $y_i^{\text{CE}}$ (from the Certainty Equivalent method) and choice frequencies $y_i^{\text{MCMC}}$ (from the MCMC method) for each gamble $z_i$ in the Marschak–Machina triangle (see Figure 2). In addition to these behavioral representations, we extracted the LLM's hidden activation at layer $l$ for each gamble $z_i$, denoted as $h_i^l \in \mathbf{R}^D$ (see Appendix A.4 for prompts). Here, $h_i^l$ is a $D$-dimensional vector, where $D$ is the latent dimension of the LLM. To align behavioral and neural representations of risk, we fit a Lasso regression to the set $\{h_i^l, y_i\}$ across all gambles by optimizing the following objective:

$$\min_{\beta_0, \beta} \{ \frac{1}{2N} \sum_{i=1}^{N} (y_i - \beta_0 - (h_i^l)^T \beta)^2 + \alpha \parallel \beta \parallel_1 \} \tag{3}$$

where $N = 1,200$ is the total number of gambles in the triangle, $\alpha$ controls the regularization strength (set to 0.1), and $\parallel \beta \parallel_1 = \sum_{j=1}^{D} |\beta_j|$ with $D$ denoting the latent dimension of the LLM.

The resulting regression coefficients, $\beta \in \mathbf{R}^D$ (also a $D$-dimensional vector corresponding to neurons in the Transformer's residual stream), are interpreted as reflecting the LLM's risk preference and are thus used as the steering vector applied to layer $l$ at inference time. In other words, this steering vector identifies the specific neural directions in the residual stream that are most predictive of the model's expressed risk preferences.

## 4 OTHER METHODS

**Contrastive Activation.** Another simple yet effective method for computing steering vectors is to contrast intermediate neural activations on carefully selected prompt pairs—a technique known as *Contrastive Activation* (Panickssery et al., 2023; Turner et al., 2023). For example, to steer LLM outputs toward more positive sentiment, CA compares the model's internal activations on a contrasting pair of prompts such as "Love" and "Hate" (Turner et al., 2023). The difference between these activations is treated as the steering vector, which is then added to the model's residual stream during inference. This shifts the model's internal representations along the desired semantic direction (e.g., toward "Love" and away from "Hate"), resulting in completions that reflect more positive sentiment.

To adapt CA to the task of steering LLM risk preferences, we constructed a list of words associated with "Risk" and "Safety" (see Appendix A.1 for details). Following the standard procedure for computing steering vectors in CA, we extracted the residual stream activations of the LLM for each word across multiple layers. The steering vector was computed as the average difference in residual activations between the risk-related and safety-related word pairs. An alternative implementation of CA could be developed by taking pairs of neural activations for gambles and/or safe options and contrasting these activations. This implementation indeed moves closer to our self-alignment methods, in which neural activations over a space of gambles are extracted from the LLM and contrasted using behavioral representations via a Lasso regression.

**Certainty Equivalent.** Finally, we consider an alternative behavioral method for eliciting individuals' risk preferences that is widely used in economics and psychology: the *Certainty Equivalent* (CE). The CE refers to the sure amount of money a person is willing to accept in place of a risky gamble (von Neumann & Morgenstern, 1947; Kahneman et al., 1979). In other words, it represents the value at which an individual is indifferent between receiving a certain payoff or a probabilistic outcome. This measure serves as a behavioral proxy for risk preference: individuals are classified as risk-averse if their CE is lower than the gamble's expected value, risk-neutral if it is equal, and risk-seeking if it is higher.

In our task, the CE serves as a direct control condition for the risk representations elicited via MCMC. Specifically, we elicited the LLM's CE for all gambles previously used in the MCMC procedure. This produces an alternative behavioral representation of risk, based on valuations rather than choices, while holding the set of gambles constant. In other words, both the CE and MCMC methods rely on the same neural representation of risk but differ in their behavioral representations.

## 5 EXPERIMENTS

In this work, we focus on steering the risk preferences of Gemma-2-9B-Instruct (Team et al., 2024), which serves as our primary target LLM. We also replicate the main experiments using Gemma-2-2B-Instruct (see Appendix F). The temperature was fixed at 1 for behavioral elicitation.

**Behavioral elicitation of risk representations.** For both the MCMC-with-LLM and CE methods, we derive steering vectors from self-aligned representations of risk. That is, these approaches rely on first eliciting the model's risk preferences through behavioral methods. To obtain a quantitative characterization of these preferences, we focus on the space of gambles defined over the probability triangle (see Figure 2). In CE, we probed Gemma-2-9B-Instruct by densely sampling gambles across the probability triangle. For each gamble, the model was prompted to report its CE. These responses were then aggregated and normalized across all sampled gambles to produce the density plot shown in Figure 2a.

Similarly, for the MCMC with LLM method, we embedded the Gemma model within a MCMC sampler, prompting it to accept or reject newly proposed gambles through binary choices. The space of gambles was identical to that used in CE (i.e., the probability triangle). The Markov chain consisted of 3,000 such binary choices. The resulting risk representation elicited via MCMC (smoothed using a Dirichlet kernel of width 0.09 that preserves probability triangle boundaries) is shown in Figure 2b. Note that the behavioral representation derived from MCMC reveals more nuanced gradients in the density plot, highlighting a stark contrast with the coarser structure observed in CE.

Steering vectors for both the MCMC and CE methods were computed using Lasso regression to align behavioral and neural representations (see Appendix B for a comparison). In contrast, the steering vector for the CA method was derived by computing the difference in neural activations between pairs of risk-related and safety-related words. To enable more effective comparisons across methods, all steering vectors were normalized by division by their Euclidean norm before applying the steering multipliers.

**Steering LLM risky choices.** Having obtained three steering vectors (derived from MCMC with LLM, CA, and CE methods), we now evaluate the effectiveness of the three steering vectors in controlling LLM's risky decision-making. Our analysis focuses on a set of four gambles that have been foundational in the study of the fourfold pattern of risk preferences (Kahneman et al., 1979). This well-documented pattern describes how human decision-makers tend to be risk-seeking when the probability of a positive outcome is low and when the probability of a negative outcome is high; conversely, they tend to be risk-averse when the probability of a positive outcome is high and when the probability of a negative outcome is low (see Table 1 for examples).

Table 1: Gambles used to evaluate the effectiveness of steering LLMs' risky decision-making. Risky options are expressed in the format {probability, outcome}; the remaining probability corresponds to receiving nothing.

|  | Outcome probability | |
|---|---|---|
|  | High | Low |
| Gains | {80%, $4000} vs $3000 | {5%, $100} vs $5 |
| Losses | {80%, -$4000} vs -$3000 | {5%, -$100} vs -$5 |

To steer the LLM's decisions toward greater risk-seeking or risk-aversion, we prompted the Gemma model with the gambles shown in Table 1, framed as binary choices between options A and B. During inference, steering vectors (scaled by a predefined multiplier ranging between -900 to +900) were added to the model's residual stream at each token position. We capped the absolute value of the multiplier at 900 because we observed unstable behaviors when using values above 1,000. The model then continued its forward pass to the output layer, where we extracted the token probabilities for "A" and "B" from the final logits (see Appendix C). These probabilities were normalized and then used to quantify the model's choice behavior under different steering conditions.

We first evaluated which Transformer layer contains the most effective residual stream for steering. To do so, we examined the model's behavior under extreme steering conditions, using multipliers of $-900$ and $+900$. For each layer, we computed the steered choice probabilities with each steering multiplier. We then quantified steerability as the average difference in choice probabilities across four gambles (see Figure 3a), defined as:

$$\text{Steerability} = \frac{1}{4} \sum_{i=1}^{4} \left( p_{\text{positive}}(z_i) - p_{\text{negative}}(z_i) \right) \tag{4}$$

where $z_i$ denotes the $i$-th gamble prompted to the LLM, $p_{\text{positive}}(z_i)$ is the model's probability of choosing the risky option under positive steering, and $p_{\text{negative}}(z_i)$ is the corresponding probability under negative steering of equal magnitude.

As shown in Figure 3b, we compared the steered choice probabilities for the risky option by subtracting the baseline (unsteered) choice probabilities. A value of zero therefore indicates no change relative to the unsteered baseline. We find that the steering vector derived from the CA method has limited impact on altering the Gemma model's choice behavior. In contrast, steering vectors computed by aligning behavioral and neural representations (i.e., both CE and MCMC methods) are effective in controlling the Gemma model's risky decision-making, shifting it toward greater risk-seeking under positive steering and greater risk aversion under negative steering.

As summarized in Table 2, the maximal ranges of steered risky choices (calculated using the optimal layer for each method) are wider for our proposed self-aligned methods (i.e., CE and MCMC) than for the CA method. This pattern is consistent across all four gambles.

**Steering LLM risk perception.** While studying abstract gambles provides valuable insights into an agent's risk preferences, psychologists have also used more naturalistic stimuli to assess people's

Table 2: Maximal range of steered risky choices for the three steering vectors (see Table 1 for the corresponding gambles). CA refers to Contrastive Activation, CE to Certainty Equivalent, and MCMC to Markov chain Monte Carlo with LLM.

| Methods | Low Probability (Gains) | High Probability (Gains) | Low Probability (Losses) | High Probability (Losses) |
|---|---|---|---|---|
| CA | 0.16 | 0.08 | 0.06 | 0.05 |
| CE | 0.93 | 0.95 | 0.90 | 0.92 |
| MCMC | 0.92 | 0.94 | 0.89 | 0.92 |

perception of risk in real-world contexts such as "cheating on an exam" or "forging someone's signature" (Weber et al., 2002; Slovic, 1987). LLMs, by virtue of their broad training data, are capable of forming meaningful risk perceptions about such real-world events (Mazeika et al., 2025; Turner et al., 2023). Indeed, recent work has shown that LLM embeddings account for a substantial portion of the variance in human risk perception (Bhatia, 2024). Here, we investigate the extent to which an LLM's risk perception can be steered using the same set of steering vectors derived in the preceding analyses.

We prompted Gemma-2-9B-Instruct to rate real-world risky events using integers from 1 (not risky at all) to 7 (extremely risky). The full set included 150 risky events curated by Bhatia (2024), spanning a range of domains: ethical (e.g., "passing off somebody else's work as your own"), financial (e.g., "betting at the horse races"), health-related (e.g., "consuming excessive amounts of alcohol"), sports (e.g., "bungee jumping"), and social (e.g., "trusting a stranger with your personal information"). As in previous experiments, we modified the residual stream during inference by injecting the steering vectors. However, instead of focusing on choices between options A and B, we extracted the model's output logits for the integer tokens "1" through "7." These token probabilities were then normalized to yield a distribution reflecting the model's perceived risk level for each event.

Analogous to the steerability metric used for risky decisions, we define steerability for risk perception as the average difference in risk ratings across all real-world events between positive and negative steering conditions. However, by comparing the most steerable layers for risky decisions (Figure 3a) and risk perceptions (Figure 4a), we find that risk perceptions are more effectively influ-

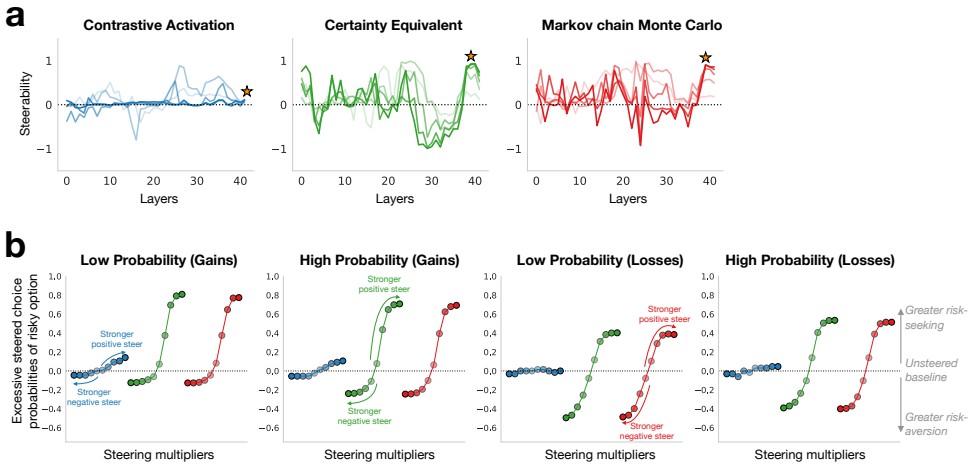

Figure 3: **Steering risky decisions of Gemma-2-9B-Instruct. (a)** Steerability results using steering vectors derived from Contrastive Activation (blue), Certainty Equivalent (green), and MCMC (red). Darker colors indicate larger steering multipliers. The optimal layers for steering, identified by the highest steerability at the maximum multiplier, are layers 41, 39, and 39 for the three methods, respectively (marked with stars). **(b)** Change in choice probabilities for the risky option after steering, using the best layer for each method. The vertical axis reflects the difference from the unsteered baseline probabilities across the four gambles.

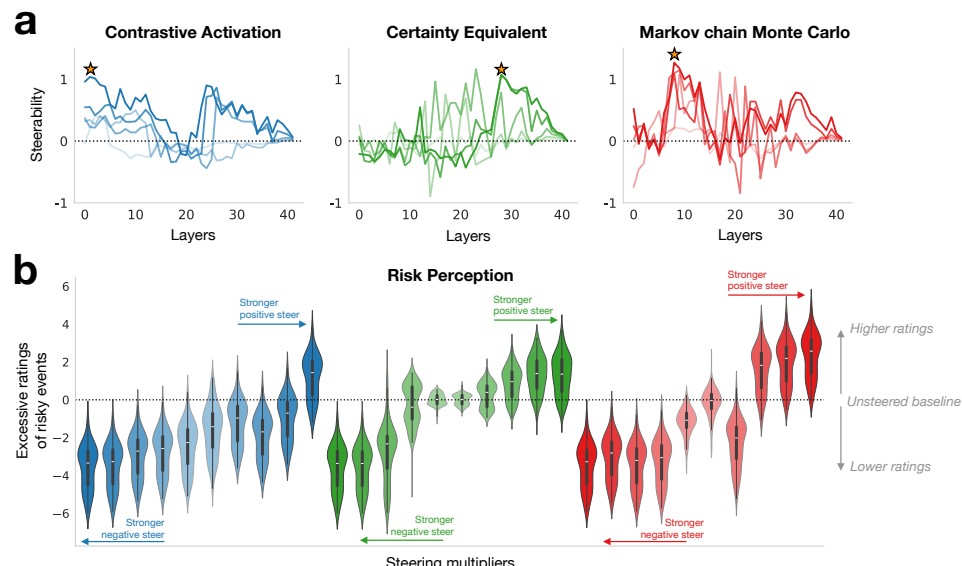

Figure 4: **Steering risk perception of Gemma-2-9B-Instruct. (a)** Steerability results using steering vectors derived from Contrastive Activation (blue), Certainty Equivalent (green), and MCMC (red). Darker colors represent larger steering multipliers. The optimal layers for steering, identified by the highest steerability at the maximum multiplier, are layers 2, 28, and 8 for the respective methods (marked with stars). **(b)** Change in average risk ratings for real-world events after steering, using the optimal layer for each method. The vertical axis reflects the deviation from the unsteered baseline rating. Each violin plot displays the distribution of ratings, with the white bar indicating the median and the black box representing the interquartile range up to the 75th percentile.

enced at earlier layers, whereas risky decisions are more steerable in later layers closer to the output of the Gemma model.

We calculated the maximal range of steered ratings, averaged across 150 real-world risky events, using the optimal layer for each method. The mean steered ranges are $4.68$ ($SD = 0.0148$) for CA, $4.93$ ($SD = 0.4211$) for CE, and $5.84$ ($SD = 0.0108$) for MCMC. These results indicate that the steering vectors derived by aligning the neural representations of gambles with the behavioral representations of risk elicited using MCMC produced the widest range of steered behavior.

Finally, we assessed responsiveness of steered ratings to the steering multiplier by computing the ratio between changes in steered ratings and changes in the multiplier. Note that, as mentioned above, all steering vectors were normalized prior to applying the steering multiplier. The mean responsiveness across the 150 events are $0.52$ ($SD = 0.0016$) for CA, $0.55$ ($SD = 0.0468$) for CE, and $0.65$ ($SD = 0.0012$) for MCMC. These results suggest that, in steering risk perceptions, MCMC is more effective, as its ratings are more responsive to changes in the multiplier.

**Steering text generation for risky events in LLM.** Besides quantitative evaluations based on choice probabilities and risk ratings, we also examine whether modifying the penultimate layer's residual stream at inference time systematically alters the textual outputs of the Gemma model. Using the same set of 150 real-world risky events described above (Bhatia, 2024), we prompted the model with the sentence "I think {event}", where {event} is replaced with each risky scenario (e.g., "I think cheating on an exam ___" or "I think consuming excessive amounts of alcohol ___"). Steering vectors were injected into the penultimate layer's residual stream at each subsequent token position during generation.

To give a visual idea of how steering influences model-generated text, we created word clouds that reflect the frequency of word usage following the application of steering vectors (see Figure 5). Overall, Gemma-2-9B-Instruct recognizes the inherent risk in the real-world events presented. However, when the residual stream at the penultimate layer is positively steered toward risk-seeking behavior using the vector derived from the MCMC method, we observe a noticeable attenuation in

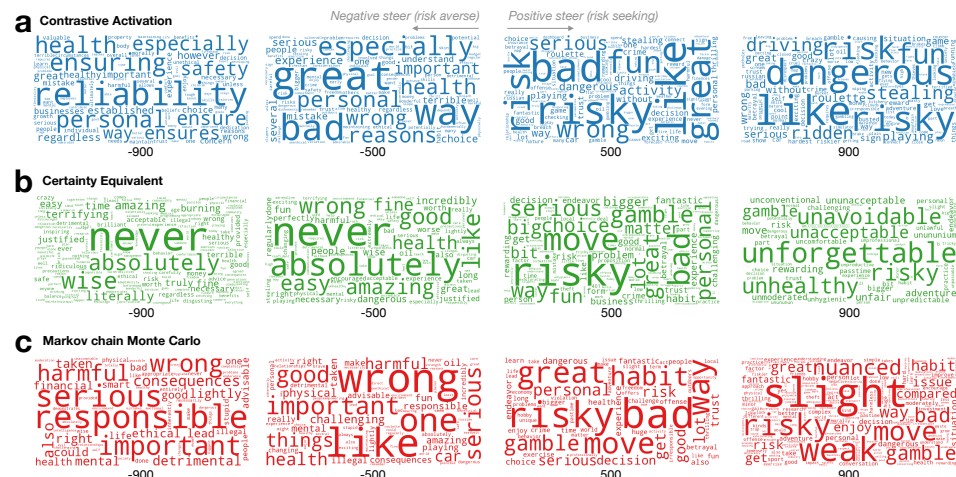

Figure 5: **Steering text generation for real-world risky events in Gemma-2-9B-Instruct.** Text outputs generated by injecting steering vectors into the residual stream at the penultimate layer of the model during inference. Steering vectors are derived from **(a)** Contrastive Activation, **(b)** Certainty Equivalent, and **(c)** MCMC with LLM methods. Each word cloud represents the frequency distribution of words used in the model's completions under different steering conditions. The corresponding steering multiplier is indicated below each word cloud.

the model's perceived risk preferences. For example, completions more frequently include phrases such as "slightly risky" and "a minor offense" (see Figure 5c). In contrast, applying the same steering vector with a negative multiplier leads the model to amplify perceived risk, generating more cautionary or morally disapproving language, such as "wrong," "never right," and "not something I would ever do." Table 3 in Appendix E presents representative examples of text completions generated by the model under different steering conditions. We also observed that injecting steering vectors at early layers of the LLM typically leads to more unstable text generations (e.g., nonsensical or gibberish text completions) compared to injections at later layers. Consistent with this finding, prior work has shown that noise injection in early layers produces only limited semantic changes, which reflects their lower levels of abstraction; however, mid-depth layers yield the most pronounced and stable deviations (Zhang et al., 2025).

**Quantitative analyses of the steered text.** To quantify the steered text generated by each method (by modifying the penultimate layer of Gemma-2-9B-Instruct), we prompted OpenAI's GPT-4.1 to judge how risk-seeking or risk-averse a person appears based on their textual description of an event (see the first prompt in Appendix A.7). GPT-4.1 returns a numerical rating between 0 (risk averse) and 100 (risk seeking) for each generated text. As shown in Figure 6a, the steered text generated using the MCMC method is more effective than that generated using CA and CE. Specifically, the mean ranges of ratings across the 150 real-world risky events are $-0.65$ ($SD = 18.87$) for CA, 3.01 ($SD = 25.15$) for CE, and 4.85 ($SD = 19.88$) for MCMC. Defining responsiveness as the ratio between changes in steered ratings and changes in the multiplier, we observe mean responsiveness values of $-0.07$ ($SD = 2.10$) for CA, 0.33 ($SD = 2.79$) for CE, and 0.54 ($SD = 2.21$) for MCMC. In sum, the MCMC method yields the greatest rating ranges and the highest responsiveness in GPT-4.1's evaluations of steered text.

Interestingly, when we instead prompt GPT-4.1 to evaluate *the riskiness of the event* based on the text descriptions–with 0 indicating "not risky at all" and 100 indicating "extremely risky" (see the second prompt in Appendix A.7)–we find that CA outperforms the self-alignment, preference-based methods (see Figure 6b). CA achieves the greatest rating range, with a mean value of 11.90 ($SD = 26.09$), compared to 6.35 ($SD = 29.81$) for CE and 6.41 ($SD = 26.39$) for MCMC. It also achieves the highest responsiveness, with a mean value of 1.32 ($SD = 2.90$), whereas CE achieves 0.71 ($SD = 3.31$) and MCMC achieves 0.71 ($SD = 2.93$). Together, these results suggest that MCMC is more effective at targeting and modifying preference-related features in text generation, whereas CA is more effective at influencing perceived riskiness.

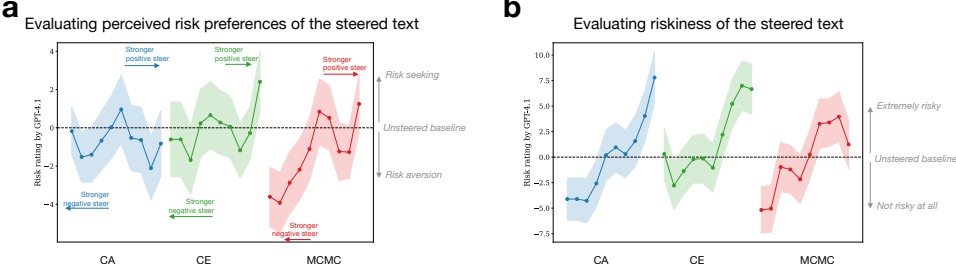

Figure 6: **Using GPT-4.1 to rate the steered text. (a)** Ratings of the perceived risk preferences of the text generator. Contrastive Activation (CA) is shown in blue, Certainty Equivalent (CE) in green, and Markov Chain Monte Carlo (MCMC) in red. **(b)** Ratings of the riskiness of the event described in the text. Shaded areas denote ± one standard error, and dots represent mean ratings.

# 6 DISCUSSION

We investigated the use of aligned behavioral and neural representations of risk to steer LLM behavior across three risk-related domains: risky decision-making, risk perception, and text generation involving real-world risky events. In all three domains, steering vectors derived from self-aligned representations (i.e., the CE and MCMC methods) consistently outperformed those generated via CA. These results suggest that self-alignment methods offer a more effective and principled means of controlling LLM behavior in risk-sensitive contexts.

**Generalizing from gambles to real-world risky events.** Both neural and behavioral representations of risk preferences were elicited using the set of three-outcome gambles defined in the Marschak–Machina triangle. Across a series of experiments, we found that the self-aligned representations provide precise control over risk in both rating tasks and text continuations involving real-world risky events. These results demonstrate that representations elicited in the domain of abstract gambles generalize effectively to more naturalistic settings. Moreover, the proposed methods can be applied to any model architecture that incorporates residual streams. The range and responsiveness of steered behavior using self-aligned steering vectors further suggest that representation engineering can provide precise behavioral control. While prompt engineering may offer a simpler alternative for altering risky behavior in LLMs, prompts often lack the nuance required to finely control model responses. Methods such as soft prompting, which optimize prompts via gradient descent, may achieve a similar level of precision, but the computational costs of identifying effective soft prompts can be substantial (Genewein et al., 2025).

**Limitations and future research.** The idea of using non-parametric methods to elicit an LLM's preferences and aligning them with corresponding neural representations should generalize to other preference domains (e.g., social or time preferences) and even to non-preference domains, provided there is a suitable stimulus space and identifiable neural representations. While self-aligned representations can intuitively steer behavior tied to the underlying construct, further theoretical and mechanistic work is needed to clarify the link between behavioral representations and neural activations. In addition, modifying the residual stream is not uniformly effective across layers, and identifying optimal layers for steering remains an open challenge. Using additional risky-choice tasks to test whether the same optimal layer is consistently identified would be helpful. Finally, steering is limited for proprietary models where internal weights or activations are inaccessible.

**Potential misuse.** The same techniques that enable fine-grained modulation of risk preferences could, in principle, be misused to steer models toward undesirable or manipulative behaviors. Because the method operates directly on LLMs' internal representations, it bypasses the transparency of prompt-based interventions, making misuse more difficult to detect. These risks highlight the need for safeguards to ensure that representation-level steering is deployed responsibly.

## ETHICS STATEMENT

Some of the real-world risky events used in our experiments include language that may be perceived as offensive. In addition, representation engineering methods such as those proposed here may be difficult to detect when deployed, raising potential concerns about transparency and misuse.

## REPRODUCIBILITY STATEMENT

We provided complete descriptions of our proposed methods in the paper. All experiments were implemented using standard Python packages from Hugging Face. The code will be released publicly upon acceptance of the paper.

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

# A PROMPTS

## A.1 WORDS FOR CONTRASTIVE ACTIVATION

Risk words: *'risk', 'uncertainty', 'danger', 'volatility', 'loss', 'gamble', 'exposure', 'threat', 'hazard', 'insecurity', 'unpredictability', 'peril', 'chance', 'vulnerability', 'instability', 'jeopardy', 'speculation', 'probability', 'accident', 'daring'*

Safe words: *'safety', 'certainty', 'stability', 'gain', 'assurance', 'protection', 'security', 'safeguard', 'reliability','predictability', 'refuge', 'guarantee', 'resilience', 'steadiness', 'shelter', 'caution', 'inevitability', 'prevention', 'prudence'*

## A.2 PROMPTS FOR CERTAINTY EQUIVALENT

*Starting with $100 in capital, what is the maximum dollar amount you are willing to pay to participate in this gamble: the gamble offers a $\{p_1\}$% chance to win $100, a $\{p_2\}$% chance to win $50, a $\{p_3\}$% chance to win $0. Respond with a single numeric value only. Do not explain your reasoning.*

## A.3 PROMPTS FOR MCMC WITH LLM

*You are participating in a gambling game where you will be shown two options, Gamble A and Gamble B:*

*Gamble A offers a $\{p_1\}$% chance to win $100, a $\{p_2\}$% chance to win $50, and a $\{p_3\}$% chance to win $0.*

*Gamble B offers a $\{p'_1\}$% chance to win $100, a $\{p'_2\}$% chance to win $50, and a $\{p'_3\}$% chance to win $0.*

*Your task is to choose between the two. Do not explain your reasoning, just limit your answer to either 'A' or 'B'.*

## A.4 PROMPTS FOR NEURAL ACTIVATIONS

*You are offered a gambling game: the gamble offers a $\{p_1\}$% chance to win $100, a $\{p_2\}$% chance to win $50, a $\{p_3\}$% chance to win $0. Respond with a single word only to express how much does this gambling appeal to you. Do not explain your reasoning.*

## A.5 PROMPTS FOR STEERING RISKY DECISION-MAKING

*Given the following gambles: Option A offers a 5% chance to win $100 and 95% chance to win $0; Option B offers a 100% chance to win $5. Please select either option A or option B to participate.*

## A.6 PROMPTS FOR STEERING RISK PERCEPTION

*Please rate how risky this behavior is with a single numeric value ranging from 1 (Not at all) to 7 (Extremely risky): $\{event\}$*

## A.7 PROMPTS FOR GPT-4.1 TO EVALUATE TEXT

**Prompting GPT-4.1 to evaluate perceived risk preference from a textual description of an event:**

*You are a helpful assistant that evaluates people's risk preference based on their opinions about an event. You will be given an opinion from a person about an event (e.g., "I think event A is . . . "). Rate how risk-seeking you think the person is on a scale from 0 to 100, where 0 means "risk aversion" and 100 means "risk seeking." Respond with only the numerical rating, without any additional text or explanation.*

**Prompting GPT-4.1 to evaluate the riskiness of a textual description of an event:**

*You are a helpful assistant that evaluates how risky a description of an event sounds. You will be given an opinion or statement about an event (e.g., "I think event A is . . . "). Rate how risky the description itself is on a scale from 0 to 100, where 0 means "not risky at all" and 100 means "extremely risky." Respond with only the numerical rating, without any additional text or explanation.*

## B  COMPARING STEERING VECTORS

In this section, we compare steering vectors derived from the MCMC and CE methods across different layers of the Gemma-2-9B-Instruct model (see Figure 7). The only difference between the two approaches lies in the behavioral representation of risk used to compute alignment; the underlying neural representation remains identical. We observe a clear trend: as we move from earlier to later layers, the similarity between the two steering vectors increases, suggesting greater convergence in their influence on model behavior at deeper levels of information processing.

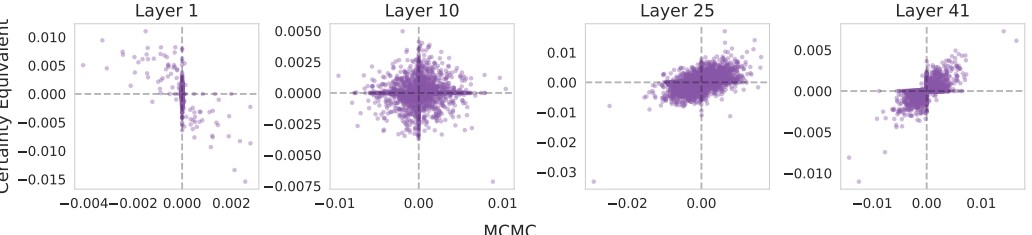

Figure 7: Comparison of steering vectors derived from MCMC with LLM (horizontal axis) and Certainty Equivalent (vertical axis) across selected layers of Gemma-2-9B-Instruct. Pearson correlation coefficients between the two steering vectors at layers 1, 10, 25, and 41 are $-0.63$ ($p < .01$), $0.04$ ($p = 0.02$), $0.48$ ($p < .01$), and $0.64$ ($p < .01$), respectively.

## C  DETAILS OF REPRESENTATION ENGINEERING

In this section, we describe the procedure for injecting a steering vector, $h_A^l$, into the residual stream of a Transformer at layer $l$. At inference time, we first obtain the model's neural activation at layer $l$ for a given prompt $p^*$:

$$h^l \leftarrow M.\texttt{forward}(p^*).\texttt{activations}[l]$$

where $M$ denotes the LLM and $h^l$ represents the original activation at layer $l$. Next, we modify this activation by injecting the steering vector scaled by a steering multiplier $c$:

$$h_S^l \leftarrow h^l + ch_A^l$$

where $h_S^l$ is the steered activation. The model then resumes its forward computation, beginning from layer $l$ with the modified activation:

$$S \leftarrow M.\texttt{continue\_forward}(h_S^l)$$

where $S$ denotes the final output generated by the steered model.

### C.1  HYPERPARAMETER VALUES

Deriving steering vectors depends on several hyperparameters, which were previously reported in a scattered manner in the main text; we reiterate them here for completeness. To elicit the behavioral representation of risk using MCMC (results shown in Figure 2b), the MCMC-with-LLM procedure generates 3000 samples. The Lasso regressions used to align behavioral and neural representations of risk employs a regularization parameter, $\alpha = 0.1$, for both self-alignment methods.

For the neural representation of risk, we enumerated all residual stream layers and injected the steering vectors at the same layer during inference for all subsequent token positions. The choice of the optimal layer is determined by the steerability metric, with the layer exhibiting the highest positive steerability selected for visualization and analysis.

## D UNDERSTANDING THE MARSCHAK-MACHINA TRIANGLE

As illustrated in Figure 2, behavioral representations of risk are defined over the Marschak–Machina triangle (Marschak, 1950; Machina, 1982), a method used to characterize preferences over three-outcome gambles. To provide additional context, we visualize the theoretical predictions of two influential models of human risky choice within this triangle. Figure 8a depicts predictions from Expected Utility Theory (EUT), a normative model of decision-making under risk (von Neumann & Morgenstern, 1947), which assumes that individuals evaluate gambles based on the weighted sum of utility. Under EUT, indifference curves are always straight and parallel, reflecting consistent trade-offs between outcomes. In contrast, Prospect Theory (PT), a descriptive model that accounts for empirical deviations from EUT, generates indifference curves that exhibit a "fanning out" pattern (see Figure 8b). This curvature reflects increasing risk aversion as the probability of extreme outcomes changes, leading to steeper indifference curves in some regions of the triangle (Machina, 1982; Harless, 1992).

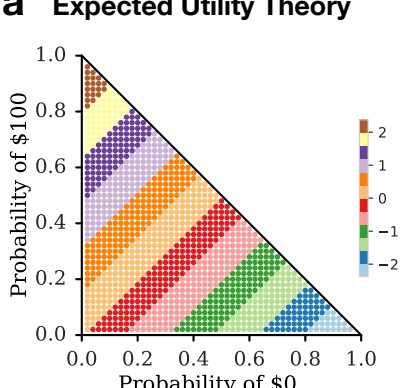 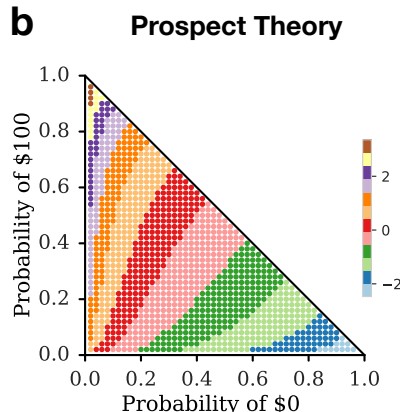

Figure 8: Predicted preferences under classical theories of human risky choice visualized in the Marschak–Machina probability triangle. **(a)** Expected utility theory (von Neumann & Morgenstern, 1947). **(b)** Prospect theory (Tversky & Kahneman, 1992) using parameters $\alpha = 0.88, \gamma = 0.52$. In both panels, each point within the triangle represents a three-outcome gamble, and color intensity reflects the model's predicted preference for that gamble. Higher values correspond to stronger preferences.

Comparing the theoretical predictions within the probability triangle to the behavioral representation elicited via MCMC (Figure 2b), we find that neither EUT nor PT qualitatively captures the risk preferences exhibited by the Gemma-2-9B-Instruct model. This suggests that existing models of human risky choice may not be directly transferable to explaining LLM behavior (Liu et al., 2024; Zhu et al., 2025). Future research should consider developing new descriptive frameworks tailored to characterizing and predicting the risky choices of LLMs.

## E EXAMPLES OF STEERING TEXT GENERATIONS

Table 3: Example steered text completions generated by Gemma-2-9B-Instruct using a steering multiplier of 900. CA refers to Contrastive Activation, CE to Certainty Equivalent, and MCMC to Markov chain Monte Carlo with LLM.

| Risky event | Steering method | Negative steer (risk averse) | Positive steer (risk seeking) |
|---|---|---|---|
| I think cheating on an exam... | CA | is wrong because it undermines the integrity of the educational system and devalues the achievements of those who uphold ethical standards. | is wrong. |
| | CE | is absolutely wrong and absolutely not worth the consequences. | is a weak move. |
| | MCMC | is wrong. | is a very bad idea. I mean, sure, maybe you might get a slightly better grade, but in the long run, it's just going to compromise your learning. |
| I think stealing an additional TV cable connection off the one you pay for... | CA | is unethical. | is stealing. |
| | CE | is not ethical and could have serious consequences. | is slightly unethical. |
| | MCMC | is not ethical. | is a slightly more minor offense compared to some other types of theft. |
| I think riding a motorcycle... | CA | is a great way to experience freedom and independence, especially for those who enjoy the outdoors and the wind in their hair. However, it's important to remember that motorcycling involves inherent risks. | is about more than just getting from point A to point B. It's about the feeling. |
| | CE | is absolutely amazing. The open air, the wind in your hair, the feeling of freedom and escape - it's truly unique. | is a rewarding but risky passtime. I enjoy the feeling of freedom and control I get while riding, but I also understand the dangers involved. |
| | MCMC | is not for everyone. It's not just about physical ability, it's also about mental and emotional preparedness. | is a very exhilarating experience. I enjoy the feeling of the wind in my hair and the freedom of the open road. |

## F    REPLICATION WITH GEMMA-2-2B-INSTRUCT

We replicated the main experiment using a smaller LLM: Gemma-2-2B-Instruct (Team et al., 2024).

**Steering risky choices.** For the four gambles presented in Table 1, Gemma-2-2B-Instruct exhibited a ceiling effect, consistently preferring the risky option with 100% choice probability. As a result, all three steering vectors showed negligible steerability in this condition.

**Steering risk perception.** Next, we examined the effects of steering vectors on the Gemma-2B model's ratings of real-world risky events (see Figure 9). The most steerable layer in the 2B model occurred at a similar relative depth as in the 9B model.

We conducted two separate two-way repeated-measures ANOVAs to evaluate the effects of steering method and multiplier on model ratings under positive and negative steering conditions.

For positive steering, the analysis revealed significant main effects of steering method, $F(2, 298) = 1710.77$, $p < .01$, and steering multiplier, $F(4, 596) = 5092.48$, $p < .01$, as well as a significant interaction between the two factors, $F(8, 1192) = 599.96$, $p < .01$. Follow-up paired $t$-tests showed that the MCMC method significantly outperformed both CE ($t(149) = 55.63$, $p < .01$) and CA ($t(149) = 35.35$, $p < .01$). Additionally, CA outperformed CE ($t(149) = 27.22$, $p < .01$).

For negative steering, we again found significant main effects of steering method, $F(2, 298) = 5786.59$, $p < .01$, and multiplier, $F(4, 596) = 2838.58$, $p < .01$, along with a significant interaction, $F(8, 1192) = 2640.75$, $p < .01$. Paired $t$-tests indicated that MCMC significantly outperformed CE ($t(149) = 100.00$, $p < .01$) and CA ($t(149) = 11.47$, $p < .01$), while CA again outperformed CE($t(149) = 89.19$, $p < .01$).

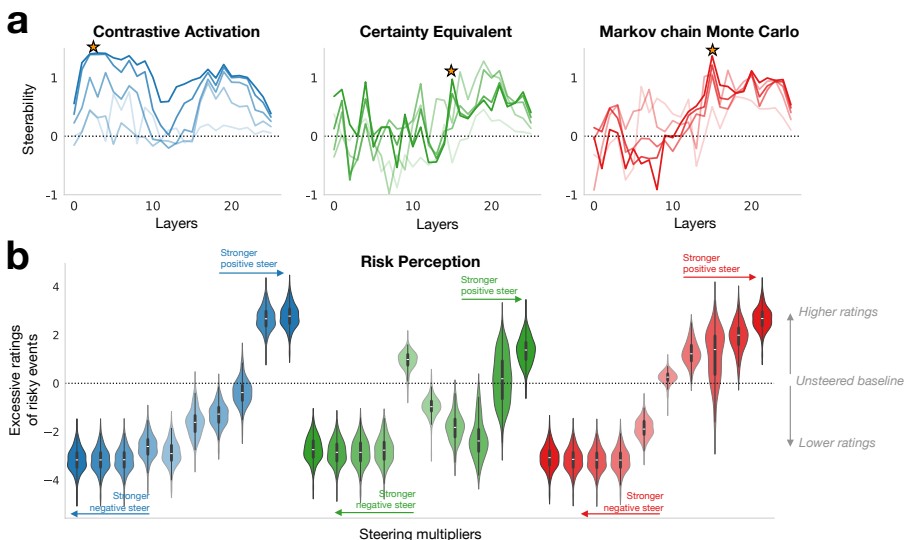

Figure 9: **Steering risk perception of Gemma-2-2B-Instruct. (a)** Steerability results using steering vectors derived from Contrastive Activation (blue), Certainty Equivalent (green), and MCMC (red). Darker colors represent larger steering multipliers. The optimal layers for steering, identified by the highest steerability at the maximum multiplier, are layers 3, 15, and 15 for the respective methods (marked with stars). **(b)** Change in average risk ratings for real-world events after steering, using the optimal layer for each method. The vertical axis reflects the deviation from the unsteered baseline rating. Each violin plot displays the distribution of ratings, with the white bar indicating the median and the black box representing the interquartile range up to the 75th percentile.

## G  REPLICATION WITH LLAMA-3.1-8B-INSTRUCT

Here, we examine the three methods for deriving steering vectors on an LLM from a different model family: Llama-3.1-8B-Instruct (Grattafiori et al., 2024). First, we elicited the Llama model's risk preferences using the CE and MCMC methods. As shown in Figure 10, the choice-based MCMC method elicited a more nuanced and structured representation of risk compared to the valuation-based CE. Based on the risk preferences elicited via the MCMC method, and similar to what we observed in Gemma-2-9B-Instruct, neither EUT nor PT provides a good account of the Llama model's risk preferences.

Comparing Figure 10b with Figure 2b, both Llama and Gemma models exhibit a fanning-out pattern in their indifference curves, suggesting that both models display increased risk aversion as the

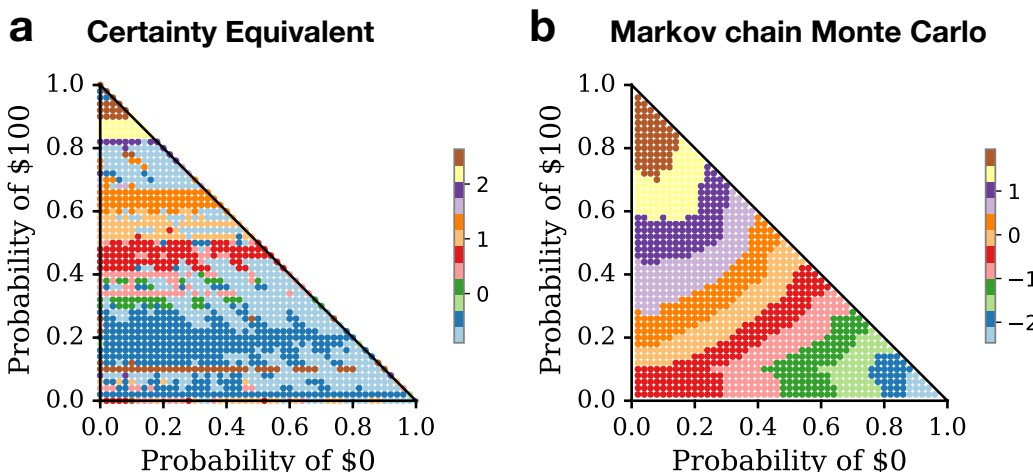

Figure 10: **Elicited risk preferences from Llama-3.1-8B-Instruct using behavioral methods. (a)** Certainty Equivalent method. **(b)** Markov chain Monte Carlo with LLM. Higher values indicate a stronger preference for the gamble by the Llama model.

probability of extreme outcomes changes. However, this fanning-out pattern is less pronounced in the Llama model than in the Gemma model.

**Steering risky choices.** We then selected the optimal layer from each method to change the choice probabilities of the four gambles in Table 1. The optimal layer for each of the three steering methods is marked with a star in Figure 11a. Table 4 reports the maximal ranges of steered risky choices, showing that the MCMC method produces a wider range of steering effects than both the CA and CE methods.

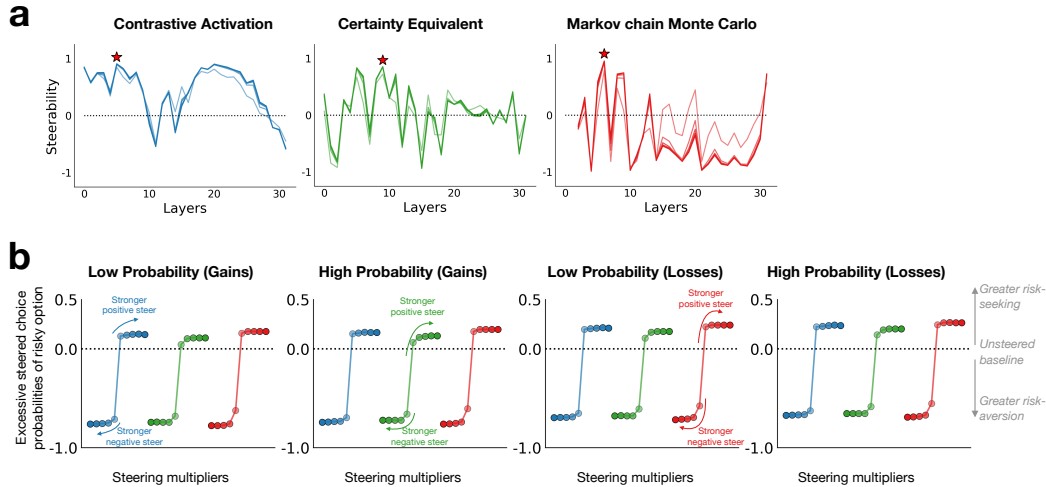

Figure 11: **Steering risky decisions of Llama-3.1-8B-Instruct. (a)** Steerability results using steering vectors derived from Contrastive Activation (blue), Certainty Equivalent (green), and MCMC (red). Darker colors indicate larger steering multipliers. The optimal layers for steering are layers 5, 9, and 6 for the three methods respectively (marked with stars). **(b)** Change in choice probabilities for the risky option after steering, using the optimal layer for each method. The vertical axis reflects the difference from the unsteered baseline probabilities across the four gambles.

Table 4: Maximal range of steered risky choices in Llama-3.1-8B-Instruct for the three steering vectors (see Table 1 for the corresponding gambles). CA refers to Contrastive Activation, CE to Certainty Equivalent, and MCMC to Markov chain Monte Carlo with LLM.

| Methods | Low Probability (Gains) | High Probability (Gains) | Low Probability (Losses) | High Probability (Losses) |
|---|---|---|---|---|
| CA | 0.91 | 0.91 | 0.91 | 0.91 |
| CE | 0.86 | 0.86 | 0.86 | 0.86 |
| MCMC | 0.95 | 0.96 | 0.96 | 0.96 |

**Steering risk perception.** Similarly, we identified the optimal layer for steering risk perception (results shown in Figure 12a). The steered ratings for these events are plotted in Figure 12b. Averaging across the same set of 150 real-world risky events curated by Bhatia (2024), the mean steered ranges are $2.42$ ($SD = 0.0136$) for CA, $3.33$ ($SD = 0.0236$) for CE, and $3.45$ ($SD = 0.1010$) for MCMC. We additionally computed the responsiveness of steered ratings to the steering multipliers. The mean responsiveness across the 150 events is $0.27$ ($SD = 0.0015$) for CA, $0.37$ ($SD = 0.0026$) for CE, and $0.38$ ($SD = 0.0122$) for MCMC. Taken together, these results indicate that the MCMC method produces wider steering ranges for real-world risky events and is more responsive to changes in the steering multiplier in Llama-3.1-8B-Instruct.

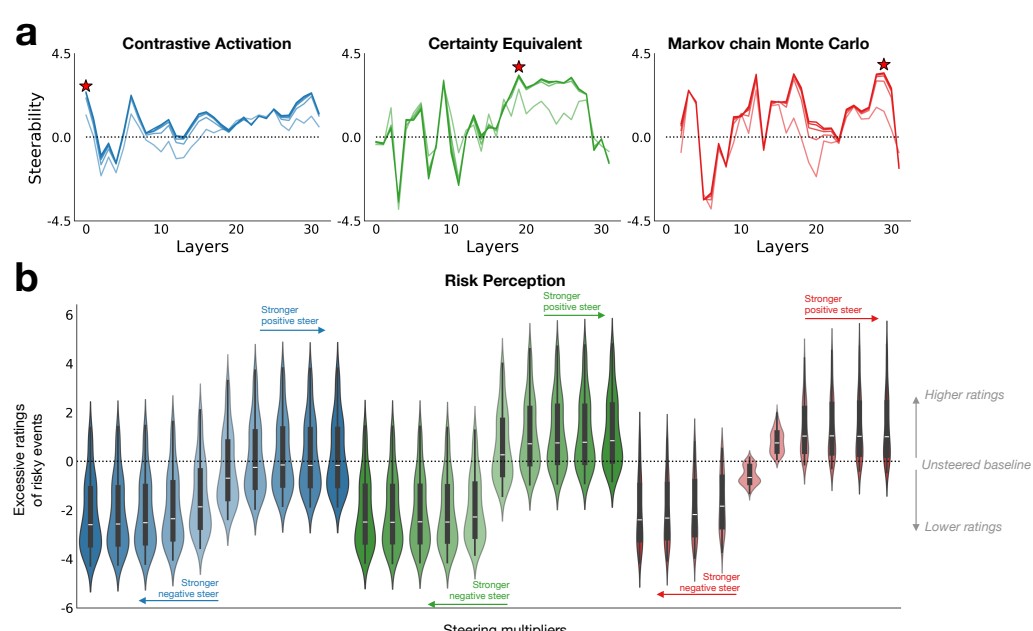

Figure 12: **Steering risk perception of Llama-3.1-8B-Instruct. (a)** Steerability results using steering vectors from Contrastive Activation (blue), Certainty Equivalent (green), and MCMC (red). Darker colors represent larger steering multipliers. The optimal layers for steering are layers 0, 19, and 29 for the respective methods (marked with stars). **(b)** Change in average risk ratings for real-world events after steering, using the optimal layer for each method. The vertical axis reflects the deviation from the unsteered baseline rating. Each violin plot displays the distribution of ratings, with the white bar indicating the median and the black box representing the interquartile range up to the 75th percentile.

## H  IMPLEMENTATION DETAILS

Steered model completions were executed on a single A100 GPU, requiring approximately 50 hours for the 9B model and 40 hours for the 2B model. Computing steering vectors via self-alignment took an additional 2 hours on a single A100 GPU.

