# OpenReview forum: "Steering Risk Preferences in Large Language Models by Aligning Behavioral and Neural Representations"
_ICLR.cc/2026/Conference — Submitted to ICLR 2026_

### Official Review · Reviewer_12bv · 2025-10-27

**Soundness:** 2
**Presentation:** 2
**Contribution:** 3
**Rating:** 4
**Confidence:** 5

**Summary:**

The paper proposes a novel “self-aligned” method for constructing steering vectors to modulate risk-related behavior in LLMs, drawing inspiration from cognitive psychology. It first elicits the model’s risk preferences using an MCMC-based choice procedure to obtain a behavioral signal, then aligns this signal with the model’s internal activations to locate the corresponding “risk” direction in residual-stream space—the steering vector. Injecting this vector at inference time shifts outputs toward risk-seeking with a positive multiplier and toward risk-aversion with a negative one.

**Strengths:**

- **Novel and more interpretable steering-vector construction:** The paper introduces a **self-aligned** steering-vector method that regresses behaviorally elicited risk preferences onto residual-stream activations, yielding a layer-specific direction. This offers clearer interpretability than mean-difference/Contrastive Activation vectors by directly tying the steering direction to the model’s revealed preferences rather than hand-picked word contrasts.
- **Principled behavioral-to-neural alignment:** The method first builds a probabilistic behavioral model of the LLM’s risk preferences via an MCMC choice process over the Marschak–Machina triangle, then aligns that distribution with neural activations using Lasso (L1=10) to extract the “risk” direction. This offers a rigorous route to extract behavior-related signals for activation engineering and avoids reliance on ad hoc prompt pairs or labels
- **Principled layer selection with a quantitative criterion:** The paper defines a steerability metric (average change between positive and negative steering) and systematically sweeps layers to pick the optimal intervention point. It also reveals a meaningful cognitive hierarchy: risky decisions steer best in later layers (e.g., layers ~39–41), while risk perception steers more effectively earlier (e.g., layers ~2–28), supporting both effective practice and interpretability about where different risk processes reside in the network

**Weaknesses:**

- In line 282~284, authors using a very large multipliers (±900) to probe most impactable layers. I am concerned that if the steered model could output plausible ansswers in this huge steering strength? since it is common that such magnitudes often risk semantic drift or degeneration in activation engineering.
- The proposed MCMC-based alignment does not clearly dominate the alternative self-aligned CE method on risky-choice control; in Table 2, CE slightly exceeds or matches MCMC on all four classic gambles.
- Only two Gamma model is evualuated in the experimental results and other commonly used model families, like Llama , Qwen, may provide more evidences on the effectiveness of the proposed method.

**Questions:**

Please see Weaknesses.

---

> ### Author Response · Authors · 2025-11-28
>
> >In line 282~284, authors using a very large multipliers (±900) to probe most impactable layers. I am concerned that if the steered model could output plausible ansswers in this huge steering strength? since it is common that such magnitudes often risk semantic drift or degeneration in activation engineering.
>
> **Responses:** Thank you for the question. To enable more effective comparisons across methods, all steering vectors were normalized by dividing by their Euclidean norm before applying the steering multipliers. Thus, a multiplier value of 900 on normalized steering vectors is not necessarily extreme for these steering tasks. However, we did observe that very large multipliers (greater than 1000) can lead to unstable responses, which is why we capped the maximum multiplier at 900. We have clarified this point in the revised paper.
>
> >The proposed MCMC-based alignment does not clearly dominate the alternative self-aligned CE method on risky-choice control; in Table 2, CE slightly exceeds or matches MCMC on all four classic gambles.
>
> **Response:** Thank you for noting the matching performance between CE and MCMC as reported in Table 2. While the range of steering differs by about 0.01 for some gambles, CE does perform slightly better on this particular task. Importantly, both CE and MCMC still outperform the non-alignment method (CA).
>
> >Only two Gamma model is evualuated in the experimental results and other commonly used model families, like Llama , Qwen, may provide more evidences on the effectiveness of the proposed method.
>
> **Response:** Thank you for this suggestion. We replicated the experiment using Llama-3.1-8B-Instruct. Please refer to the new Section G, “Replication with Llama-3.1-8B-Instruct,” in the Appendix for details. We find qualitatively similar results using the Llama model as those obtained with the Gemma model.

---

### Official Review · Reviewer_ypLS · 2025-10-30

**Soundness:** 3
**Presentation:** 3
**Contribution:** 3
**Rating:** 6
**Confidence:** 4

**Summary:**

The paper proposes a steering methodology termed "self-alignment" where steering vectors are determined by aligning model's latent behavioral representations (gamble scores etc.) and neural representations (Transformer residual stream activations). Authors specifically focus on controlling risk-seeking behavior of LLMs.

**Strengths:**

1. **Novelty:** Adaptation of the MCMC procedure from Noguchi et. al. (2013) by replacing people with an LLM is an interesting touch with its application in LLM steering.
2. **Originality:** Estimating the steering vector from models own preference without external datasets of opposing prompts is not very common (to my knowledge).
3. **Significance:** The techniques (in Step 1 and 2), even though individually not brand new, adopted for LLMs can be useful contribution to the ICLR community working on AI alignment.

**Weaknesses:**

1. **Writing:** Although the paper is well-written and presents an easy-to-follow narrative, Section 3 reads with some friction, as most mathematical objects are described verbally rather than symbolically. Explicitly casting the output samples from Step 1 into mathematical variables, passing them into Step 2, and formally expressing the lasso regression problem would lower the cognitive load required from readers to understand the method.
2. **Generality:** The prompt set construction method (random sequence of gambles) is directly inherited from Noguchi et al (2013). This makes the method inherently tailored for risk preference modeling only. While this aligns with the paper's stated goal, it also limits the method's applicability beyond that specific domain (at least not as easily extendible to most type of preferences as Contrastive Activation Addition). This might restrict the independent contribution of the paper to just replacing the human feedback with LLM responses.

**Questions:**

1. **Generality:** Could authors comment on potential principles to extend the methodology for steering different types of preferences?
2. **Transferability:** Although it is the easiest question to ask, could authors provide any clue if the proposed self-alignment steering method works well with different model families (e.g. Qwen, Llama)?

___

Overall, I believe the paper delivers on what it promises and the adaptation of a historical methodology seems interesting and useful for advancing LLM behavior control research.

---

> ### Author Response · Authors · 2025-11-28
>
> >Writing: Although the paper is well-written and presents an easy-to-follow narrative, Section 3 reads with some friction, as most mathematical objects are described verbally rather than symbolically. Explicitly casting the output samples from Step 1 into mathematical variables, passing them into Step 2, and formally expressing the lasso regression problem would lower the cognitive load required from readers to understand the method.
>
> **Response:** Thank you for this suggestion. We have expanded the alignment section and explicitly spelled out the optimization objective of the Lasso regression. Please refer to the revised section, “Step 2: Aligning behavioral and neural representations to compute steering vectors” for more details.
>
> >Generality: The prompt set construction method (random sequence of gambles) is directly inherited from Noguchi et al (2013). This makes the method inherently tailored for risk preference modeling only. While this aligns with the paper's stated goal, it also limits the method's applicability beyond that specific domain (at least not as easily extendible to most type of preferences as Contrastive Activation Addition). This might restrict the independent contribution of the paper to just replacing the human feedback with LLM responses.
>
> >Generality: Could authors comment on potential principles to extend the methodology for steering different types of preferences?
>
> **Response:** Thank you for the comment and the question. The general principles behind our self-alignment methods include: (i) using non-parametric methods, such as MCMC-with-LLM, to elicit an LLM’s preferences, and (ii) aligning these behaviorally elicited representations with the corresponding neural representations. These principles should generalize to other types of preferences, such as social preferences or time preferences, provided that the underlying domain of stimuli is quantifiable and can be sampled from (as gambles are for risk preferences). We expect these preferences to extend naturally to more realistic settings that involve the same underlying preference structures. We have added discussion of this point to the revised paper.
>
> >Transferability: Although it is the easiest question to ask, could authors provide any clue if the proposed self-alignment steering method works well with different model families (e.g. Qwen, Llama)?
>
> **Response:** Thank you for this suggestion. We replicated the experiment using Llama-3.1-8B-Instruct. Please refer to the new Section G, “Replication with Llama-3.1-8B-Instruct,” in the Appendix for details. We find qualitatively similar results using the Llama model as those obtained with the Gemma model.

---

### Official Review · Reviewer_Mr6i · 2025-10-31

**Soundness:** 2
**Presentation:** 2
**Contribution:** 2
**Rating:** 4
**Confidence:** 3

**Summary:**

This paper proposes a self-alignment method to achieve precise control over risk-related behaviors in LLMs. Through the proposed MCMCb-based method, the latent risk preference representations of the LLM are inferred from its behavioral choices. Then Lasso regression is employed to compute a steering vector that aligns behavioral and neural representations. By injecting the steering vector into the model’s residual stream at each token position, the risk-related decision-making of the model can be effectively controlled.

**Strengths:**

1. The paper proposes a new self-alignment method that uniquely combines the probability triangle, MCMC method, and lasso regression to manipulate risk preferences in large language models.

2. The experimental evaluation is extensive, covering three different but related tasks: risky decision-making, risk perception, and text generation.

3. The method uses the model itself to identify steering vectors associated with risk preferences, showing great potential for practical applications.

**Weaknesses:**

1. The experiments are conducted only on limited Gemma-models. It is unclear whether the same performance improvement can be achieved on other large-scale models.

2. Lack of the key details. During the construction of the steering vector, the paper does not explain how the Lasso regularization coefficient, MCMC sampling steps, or injection layer selection were chosen.

3. High initialization cost. When switching to another model or modifying other attributes, the steering process must start from scratch—although inference-time cost is low, the initialization cost is high.

**Questions:**

1. Could the authors conduct additional experiments on other open-source large language models to further verify the effectiveness and generalizability of the proposed method?

2. Could the authors incorporate additional risky-choice tasks to examine whether the same “optimal layer” remains consistent across different categories of risk-related decision-making tasks?

3.  Could the authors clarify whether the construction of risk-related steering vectors can be made deterministic and fully guaranteed when derived from gamble-based task? If add more risk-related task can improve the final performance?

4. It would strengthen the paper if the authors could include ablation experiments to quantify the contribution and importance of each component in the proposed framework.

5. The explanation of Step 2 in Section 3 could be expanded with a more detailed and rigorous description of how Lasso regression is used to align behavioral and neural representations.

---

> ### Author Response · Authors · 2025-11-28
>
> >The experiments are conducted only on limited Gemma-models. It is unclear whether the same performance improvement can be achieved on other large-scale models.
>
> >Could the authors conduct additional experiments on other open-source large language models to further verify the effectiveness and generalizability of the proposed method?
>
> **Response:** Thank you for the comment and the question. We replicated the experiment using Llama-3.1-8B-Instruct. Please refer to the new Section G, “Replication with Llama-3.1-8B-Instruct,” in the Appendix for details. We find qualitatively similar results using the Llama model as those obtained with the Gemma model.
>
> >Lack of the key details. During the construction of the steering vector, the paper does not explain how the Lasso regularization coefficient, MCMC sampling steps, or injection layer selection were chosen.
>
> **Response:** Thank you! Some hyperparameter values were reported in the main text but in a scattered manner. We now put them altogether in the new Appendix Section C.1.
>
> >High initialization cost. When switching to another model or modifying other attributes, the steering process must start from scratch—although inference-time cost is low, the initialization cost is high.
>
> **Response:** Thank you for raising the concerns about the computational cost of the proposed methods. The elicitation of neural representations of gambles is shared between both self-alignment methods (i.e., CE and MCMC), so that portion of the computation is identical. Thus, the primary differences in compute arise from the behavioral elicitation of risk preferences.
>
> For the domain of gambles, the computational cost is manageable within the probability triangle (e.g., see Figure 2), as most of the compute is spent on eliciting choices or valuations from the LLMs. We agree that if the gamble space, or the space of other relevant attributes, becomes non-enumerable, the computational cost could become substantial. However, there is value in the MCMC sampling approach, which can greatly simplify behavioral elicitation in higher-dimensional spaces.
>
> >Could the authors incorporate additional risky-choice tasks to examine whether the same “optimal layer” remains consistent across different categories of risk-related decision-making tasks?
>
> **Response:** Thank you for the question. We would like to clarify that both the CE and MCMC methods already involve many risky-choice tasks, as the probability triangle spans a wide range of possible gambles. If the reviewer is suggesting the use of non-gamble risky-choice tasks (for example, risky decisions involving more naturalistic stimuli such as those used in Marantz & Plonsky, 2025, arXiv:2503.14004), we agree that such an experiment would be valuable (though outside the scope of the current paper). We include a discussion on how this type of analysis could help assess the consistency of the optimal layer across different classes of risky-choice tasks in the revised paper.
>
> >Could the authors clarify whether the construction of risk-related steering vectors can be made deterministic and fully guaranteed when derived from gamble-based task? If add more risk-related task can improve the final performance?
>
> **Response:** The neural representations of gambles are fixed across the Marschak–Machina probability triangle. Because of the large sample size, the MCMC-with-LLM method also covers the triangle well. Together, these factors allow the regression to derive relatively stable steering vectors. However, in more complex domains, the behavioral and neural representations of a preference may not be as stable, especially if the MCMC sample size is small. In such cases, the derived steering vectors may vary. It is also possible that the LLM may not exhibit stable preferences at all, in which case we would not expect the self-alignment method to function effectively. While we could expand the space of gambles, we do not expect that adding additional risk-related tasks would substantially improve the quality of the steering vectors.
>
> >It would strengthen the paper if the authors could include ablation experiments to quantify the contribution and importance of each component in the proposed framework.
>
> **Response:** Thank you for the comment. We have implemented several control experiments. Between the CE (valuation-based self-alignment) and MCMC (choice-based self-alignment) methods, the neural representations of gambles are identical, so any difference in the effectiveness of steering vectors can be attributed to the behavioral representation of risk. Our results show that choice-based behavioral representations perform better. Moreover, between the CA (non-alignment) method and the self-alignment methods (including both CE and MCMC), we test whether aligning the behavioral and neural representations is more effective than deriving steering vectors solely from neural representations.

---

> > ### Author Response · Authors · 2025-11-28
> >
> > >The explanation of Step 2 in Section 3 could be expanded with a more detailed and rigorous description of how Lasso regression is used to align behavioral and neural representations.
> >
> > **Response:** Thank you for the comment. We have expanded that section with a more detailed description of the Lasso regression. The newly added Equation 3 now explicitly presents the optimization objective along with the corresponding hyperparameter values.

---

### Official Review · Reviewer_4JjW · 2025-10-31

**Soundness:** 2
**Presentation:** 3
**Contribution:** 2
**Rating:** 2
**Confidence:** 4

**Summary:**

The paper presents a novel approach to identifying steering vectors in an LLM via behavioral preference tests, specifically in the domain of risk preferences, and then successfully steers the model with those vectors on risk-related prompts. It first elicits granular behavioral preferences using methodology derived from the human psychology literature, then uses regression to identify activation-space predictors of those preferences, which are used as the steering vector.

**Strengths:**

The approach is novel, and a persuasive case is made that the MCMC method is the right way to capture the structure of risk preferences in an LLM.

**Weaknesses:**

The much simpler Contrastive Activation method is not offered a fair comparison. The paper's contrast of "risk" and "safety" related words would have induced a vector related to the abstract concept of risk, but the behavioral methods identify vectors related to quantitative risk preferences. Thus when steering on risk preference-related prompts (Figure 3), the former is ineffective. A more appropriate comparison would be to a vector formed by contrasting risky with safe choices. The paper's contrast vector is more appropriate for steering on risk perception-related prompts (Figure 4), and there it appears to be similarly effective to the more compute-intensive methods. The vector is also ill-suited to inducing "risk seeking" (Figure 5); again, a vector elicited by contrasts over preferences would be more appropriate.

It's very difficult to draw conclusions of any sort from the word clouds in Figure 5.

It's not clear how well this methodology would extend beyond risk preferences, nor whether it could even be applied in non-preference-related domains.

**Questions:**

Steering vectors of magnitude 900 on unit-normed vectors are shockingly high; I don't recall ever seeing that in the literature. Why was that magnitude chosen? How does it compare to the activation magnitudes pre-steering? Were outputs not degraded with such extreme magnitudes?

Where might this approach be applied in practice? Contrastive activations are simple and general; does this approach offer any benefit beyond steering risk preferences?

---

> ### Author Response · Authors · 2025-11-28
>
> >The much simpler Contrastive Activation method is not offered a fair comparison. The paper's contrast of "risk" and "safety" related words would have induced a vector related to the abstract concept of risk, but the behavioral methods identify vectors related to quantitative risk preferences. Thus when steering on risk preference-related prompts (Figure 3), the former is ineffective. A more appropriate comparison would be to a vector formed by contrasting risky with safe choices. The paper's contrast vector is more appropriate for steering on risk perception-related prompts (Figure 4), and there it appears to be similarly effective to the more compute-intensive methods. The vector is also ill-suited to inducing "risk seeking" (Figure 5); again, a vector elicited by contrasts over preferences would be more appropriate.
>
> **Response:** Thank you for raising this interesting suggestion. Our interpretation of the Contrastive Activation (CA) method is that it takes contrasting pairs of words and computes the difference in their neural activations. However, as the reviewer suggests, one could analogously take a pair of gambles (e.g., a risky option vs. a safe option), extract the neural activations for each gamble, and compute their difference. This idea indeed moves closer to our self-alignment approach.
>
> In our method, we elicit neural representations for all gambles in the Marschak–Machina probability triangle. These neural representations are then contrasted and reweighted according to the LLM’s preferences over gambles (elicited using MCMC-with-LLM) via the Lasso regression. In this sense, the self-alignment method can be viewed as a more sophisticated reweighting scheme for CA within the triangle space of gambles: behaviorally elicited risk preferences serve as guiding signals for how to compare and contrast the neural representations of each gamble.
>
> We have added relevant discussion about this potential connection in the revised paper.
>
> >It's very difficult to draw conclusions of any sort from the word clouds in Figure 5.
>
> **Response:** Thank you for this comment. We conducted a new analysis using OpenAI’s GPT-4.1 to evaluate steered textual descriptions of real-world risky events. We tested two variants: (1) GPT-4.1 rating the degree of risk-seeking or risk aversion expressed in the text, and (2) GPT-4.1 rating the degree of riskiness conveyed by the text. The results are presented in the newly added Figure 6a (for variant 1) and Figure 6b (for variant 2).
>
> The MCMC method exhibits the greatest range and responsiveness in ratings of perceived risk preferences of the text generator. Interestingly, the CA method shows the greatest range and responsiveness in ratings of the riskiness expressed in the text. Taken together, these results suggest that the self-alignment methods are more effective at targeting preference-related features of text, whereas CA is more effective at targeting risk-related features of text.
>
> Further details are provided in the new section titled “Quantitative analyses of the steered text.”
>
> >It's not clear how well this methodology would extend beyond risk preferences, nor whether it could even be applied in non-preference-related domains.
> Where might this approach be applied in practice? Contrastive activations are simple and general; does this approach offer any benefit beyond steering risk preferences?
>
> **Response:** Thank you for the comment and the question. The general principles behind our self-alignment methods include: (i) using non-parametric methods, such as MCMC-with-LLM, to elicit an LLM’s preferences, and (ii) aligning these behaviorally elicited representations with the corresponding neural representations. These principles should generalize to other types of preferences, such as social or time preferences, provided that the underlying stimulus domain is quantifiable and can be sampled from (as gambles are for risk preferences). We expect these preferences to extend naturally to more realistic settings that involve the same underlying preference structures.
>
> We also believe these principles can generalize to non-preference domains, as long as there exists a stimulus space from which samples can be drawn to elicit behavioral representations and a corresponding neural representation can be identified.
>
> We have added discussion of these points to the paper.

---

> > ### Author Response · Authors · 2025-11-28
> >
> > >Steering vectors of magnitude 900 on unit-normed vectors are shockingly high; I don't recall ever seeing that in the literature. Why was that magnitude chosen? How does it compare to the activation magnitudes pre-steering? Were outputs not degraded with such extreme magnitudes?
> >
> > **Responses:** Thank you for the question. To enable more effective comparisons across methods, all steering vectors were normalized by dividing by their Euclidean norm before applying the steering multipliers. Thus, a multiplier value of 900 on normalized steering vectors is not necessarily extreme for these steering tasks. However, we did observe that very large multipliers (greater than 1000) can lead to unstable responses, which is why we capped the maximum multiplier at 900. We have clarified this point in the paper.

---

### Author Response · Authors · 2025-12-02

Dear AC,

To assist the final decision, we provide a summary of our work and the changes implemented during the rebuttal period.

Our work has a clear goal: to propose a principled method for identifying steering vectors that, when injected at inference time, systematically alter an LLM’s risk preferences. We show that aligning an LLM’s own emergent behavioral representations of risk preference with its neural representations (hence the term **self-alignment**) is effective for discovering such vectors. Although the alignment is learned using gambles commonly used in the lab to measure human risk preferences, the resulting steering vectors generalize well to more naturalistic settings, including judging the riskiness of real-world events and altering LLM text generation when describing those events.

Three of the initial reviews are positive or borderline (scores 6, 4, and 4). They note that “the experimental evaluation is extensive, covering three different but related tasks,” that “the method uses the model itself to identify steering vectors associated with risk preferences, showing great potential for practical applications” (Mr6i), that “originality: estimating the steering vector from the model’s own preference without external datasets of opposing prompts is not very common” (ypLS), and that the method provides a “novel and more interpretable steering-vector construction” with “principled behavioral-to-neural alignment” (12bv). Even the more critical reviewer (score 2) acknowledged the novelty of the approach, writing that “the approach is novel, and a persuasive case is made that the MCMC method is the right way to capture the structure of risk preferences in an LLM” (4JjW).

The main concerns raised by the reviewers who scored 2 or 4 were:

(1) Although the original submission evaluated two Gemma models, it would be useful to test an additional open-weight LLM from another model family;

(2) The need for a quantitative evaluation of the steered text generation.

We directly addressed both points. We conducted additional experiments using Llama-3.1-8B-Instruct and replicated our main findings (see the new Appendix G). To quantify steered text, we prompted GPT-4.1 to judge the perceived risk preferences expressed in the steered textual outputs. These analyses show that self-aligned methods are more effective for this task (see the new section “Quantitative analyses of the steered text”).

We have also addressed all remaining minor comments, with changes marked in blue. We thank the reviewers for their time and thoughtful feedback, and we refer the AC to the detailed discussion below for further clarifications and context.

Kind regards,

The Authors

---

### Meta-Review · Area_Chair_Q2bE · 2026-01-08

**Summary:**

The paper proposes a self-alignment approach for constructing steering vectors to control risk preferences in large language models by aligning behaviorally elicited representations with internal neural activations. While reviewers acknowledge that the problem setting is interesting and that the paper draws inspiration from cognitive psychology in a thoughtful way, they raised substantial concerns regarding the strength and generality of the contribution. In particular, reviewers found that the method’s comparison and discussion relative to existing steering and contrastive activation approaches is limited, the empirical gains over simpler baselines are not consistently compelling, and the scope of applicability appears narrowly tailored to risk preference modeling. Taken together, this paper is not recommended for acceptance at its current form.

**Reviewer Concerns:**

The rebuttal addressed several concerns, including adding experiments on an additional model family, clarifying hyperparameter choices, and introducing a quantitative analysis of steered text generation. While these additions improve completeness, they do not fully resolve the core issues raised by reviewers. Concerns remain about whether the proposed self-alignment method meaningfully outperforms simpler or less compute-intensive alternatives. Reviewers also expressed ongoing doubts about the reliance on large steering magnitudes, the lack of strong ablations isolating the benefit of the MCMC-based alignment, and the limited evidence that the approach generalizes beyond risk preferences to broader behavioral or alignment settings. As a result, the rebuttal did not substantially change reviewers’ assessments of the paper’s technical depth, robustness, or broader impact.

**Reviewer Scores:**

Given that the central concerns regarding generality and empirical justification remain unresolved, no upward score changes are expected.

---

### Decision · Program_Chairs · 2026-01-26

Reject